# *Arthrobotrys mendozadegivensis* sp. nov. (Fungi: Orbiliales) from Mexico: Predatory Activity and Nematocidal Activity of Its Liquid Culture Filtrates Against *Haemonchus contortus* (Nematoda: Trichostrongylidae)

**DOI:** 10.3390/jof10120888

**Published:** 2024-12-22

**Authors:** Enrique Gutiérrez-Medina, Pedro Mendoza-de Gives, Gustavo Pérez-Anzúrez, Antonio Colinas-Picazo, Génesis Andrea Bautista-García, Miguel Ángel Alonso-Díaz, Elke von Son-de Fernex, María Eugenia López-Arellano

**Affiliations:** 1Laboratory of Helminthology, National Centre for Disciplinary Research in Animal Health and Innocuity (CENID-SAI), National Institute for Research in Forestry, Agriculture and Livestock (INIFAP-AGRICULTURA), Jiutepec 62550, Mexico; egm.20797@gmail.com (E.G.-M.); tavopzaz@gmail.com (G.P.-A.); antoniocolinasp@gmail.com (A.C.-P.); bagg150583@gmail.com (G.A.B.-G.); mlopez.arellano@gmail.com (M.E.L.-A.); 2Production Sciences and Animal Health, Faculty of Veterinary Medicine and Zootechnics, National Autonomous University of Mexico (UNAM), Coyoacán, Mexico City 04510, Mexico; 3Tropical Livestock Center, Faculty of Veterinary Medicine and Zootechnics, National Autonomous University of Mexico, Martínez de la Torre 93600, Mexico; alonsodma@hotmail.com (M.Á.A.-D.); elkevsdf@comunidad.unam.mx (E.v.S.-d.F.)

**Keywords:** *Arthrobotrys*, sp. nov., predation, orbiliales, nematophagous fungi, taxonomic identification, Mexico

## Abstract

During the isolation, identification, and assessment of nematode-trapping fungi (NTF) against nematodes, we discovered an unusual fungus in decaying wood from Morelos State, Mexico. This isolate exhibited some characteristics similar to those of the *Arthrobotrys* genus; however, we found that it did not match any previously reported species within this genus after conducting morphological and phylogenetic analyses using the ITS, TEF, and RPB2 regions. This new species displays conidiophores with two or three stems emerging from the same initial site and conidiophores with only a single stem and aerial thickened hyphae from which single conidiophores emerge, forming 3D adhesive nets. The conidia, which have one or two septa, range from obovoid to ellipsoidal, crowned by four to six conidia. This report provides evidence that this species has not been described before, and we hereby introduce it as a new species, naming it *Arthrobotrys mendozadegivensis*. This species displayed a predatory activity of 76.92%, and its liquid culture filtrates in Sweet Potato Dextrose Broth and Czapek–Dox Broth were effective in killing 40.90% and 34.91% of *Haemonchus contortus* larvae, respectively. This study provides information about a previously unreported species of nematophagous fungus, which is important for systematics and has potential biotechnological applications against nematodes that affect the livestock industry.

## 1. Introduction

Nematode-trapping fungi (NTF) are a group of natural enemies of nematodes that inhabit soil, forming a regular part of the soil mycobiota. These fungi can be found in various environments worldwide, including tropical regions [1,2], high plateaus [3], forests [4], mangroves [5], deciduous woodlands [6], marine ecosystems [7], freshwater [8], and even the Antarctic [9]. This group of micro-fungi has existed for millions of years [10]. NTF are classified into three major groups: Zoopagomycota (Zoopagales), Basidiomycota (Polyporales and Agaricales), and Ascomycota (Orbiliomycetes). Within the Orbiliomycetes class, different species exhibit various strategies to capture, kill, and feed on nematodes. Some genera and species utilize adhesive and mechanical trapping methods, including three-dimensional structures, simple or constricting rings, and adhesive columns or buttons [11]. This important group of fungi plays a crucial ecological role in nature as decomposers of organic matter, contributes to nitrogen cycling in soil, is involved in food chains, and acts as a significant regulator of nematode populations [12]. The genus *Arthrobotrys* belongs to the Orbiliomycetes and was first described in 1839 [13]. Later, Zopf [14] referred to this fungus as a carnivorous species that preys on nematodes through the production of trapping devices [15]. As of now, more than a hundred species within the *Arthrobotrys* genus have been identified [16]. These fungi produce tiny, waxy, translucent, light-colored apothecia that can be sessile or sub-stipitate. Members of the Orbiliaceae family possess small ascospores that are asymmetrically globose to sub-fusoid [17]. The present study describes a new species of NTF isolated from decaying wood in Coatlán del Río Municipality, Morelos State, Mexico, based on its morphological characteristics and molecular analysis using the ITS, TEF, and RPB2 amplified regions via PCR, following protocols by White et al. [18], Stielow et al. [19], and Zhang et al. [16]. After thorough morphological and molecular analysis, this fungus exhibited characteristics similar to those of the *Arthrobotrys* genus. However, it did not fully align with any previously reported *Arthrobotrys* species, prompting the authors to designate this isolate as a new species, naming it *Arthrobotrys mendozadegivensis*. Additionally, this study evaluated the predatory activity of this fungus and the nematocidal activity of its liquid culture filtrates (LCF) against the sheep parasitic nematode *Haemonchus contortus.* This new species of NTF could have significant implications for future research as a potential biotechnological tool for controlling parasitic nematodes that are important in agriculture and livestock management.

## 2. Materials and Methods

### 2.1. Fungal Isolation

Fifty grams of decaying wood was collected from the Eco-Touristic Park “El Hoyanco” in the Municipality of Coatlán del Río, located in the State of Morelos, Mexico. Coatlán del Río is situated in the southwestern part of Morelos. This area is characterized by deciduous forests, with an average temperature of 24 °C and an annual rainfall of 1000 mm (Figure 1).

One gram of decaying wood was crushed into small particles and sprinkled on the surface of sterile water agar plates (90 × 15 mm). These plates were then incubated for 7 days at room temperature (18–25 °C). After this initial incubation, a few drops of an aqueous suspension containing an undetermined number of specimens of the free-living nematode *Panagrellus redivivus* were added to the agar surface as bait to stimulate the growth of aerial structures of nematode-trapping fungi [20]. Four days after introducing the nematodes, fungal aerial structures began to develop on the agar surface. These structures included conidiophores, conidia, and trapped nematodes. Aerial conidia were collected with a sterile needle and transferred to new sterile water agar plates. These fresh plates were incubated at the same temperature and examined under a microscope twice a week. To isolate the fungi, we propagated them on sterile water agar plates through successive monoconidial transfers until we obtained pure cultures [21] (Figure 2).

### 2.2. Morphological Identification

The microculture technique was carried out as follows: A small square block (1 cm^2^) was cut from a sterile potato–dextrose agar plate and placed on a slide [22]. This slide, along with the agar block, was positioned in a Petri dish (100 mm diameter). The agar block was inoculated on all four sides with the fungus and then covered with a coverslip. A thin film of sterile water was added to the bottom of the Petri dish to create a humid chamber [23]. The microcultures were incubated for 7 days at room temperature (18–25 °C). The Petri dishes were sealed with parafilm during this period. After 7 days, the coverslips containing the fungal growth were removed from the microcultures and placed over a water droplet on a slide. The fungal structures were then observed under a microscope at 40× and 100× magnifications. Additionally, water agar plates were inoculated with the mycelium of the isolate under study. Water droplets containing an undetermined number of specimens of *P. redivivus* were added, and the plates were allowed to incubate for another 7 days at the same room temperature. Free-living nematodes were commercially acquired as living fish food from a local pet store. Fungal structures including conidiophores, conidia, trapping devices, chlamydospores, and trapped nematodes were observed and analyzed. The lengths and widths of the conidia, as well as the lengths of the conidiophores, were measured using a Leica Zeiss DM6B microscope (Wetzlar, Germany). A total of 25 structures (i.e., conidia and conidiophores) were randomly selected for measurement. Descriptions of the main taxonomically important structures were recorded, and a series of microscopic images of these structures were captured using the camera of the same microscope, employing the LAS V4.9 program [23].

### 2.3. Molecular Identification

#### Genomic DNA Isolation

The genomic DNA from this isolate was extracted using the Wizard^®^ Genomic DNA Purification Kit (PROMEGA, Madison, WI, USA). The concentration of the isolated genomic material was measured using a NanoPhotometer NP80 (IMPLEN, Munich, Germany). The extracted DNA was processed through endpoint PCR by following the protocols outlined by Tigano-Milani et al. (1995) [24], and these methods were standardized in the Helminthology Laboratory of INIFAP. The PCR reaction was performed in a final volume of 20 µL, which included 100 ng of genomic DNA, 10 µL of GoTaq^®^ Green Master Mix 2X (PROMEGA, Madison, WI, USA), and 1.5 µL of each oligonucleotide at a concentration of 20 µM: ITS5-fw (5′-TCCTCCGCTTATTGATATGC-3′) and ITS4-rv (5′-GGAAGTAAAAGTCGTAACAAGG-3′) for the ITS region [18]; EF-1018F (GAYTTCATCAAGAACATGAT) and EF-1620R (GACGTTGAADCCRA-CRTTGTC) for the TEF1-α region [19]; and RPB2-6f (TGGGGKWTGGTYTGYCCTGC) and fRPB2-7f (CCCATRGCTTGYTTRCCCAT) for the RPB2 region [16]. Nuclease-free water was added to complete the reaction volume of 20 µL. The amplification conditions for the ITS region were as follows: an initial denaturation at 94 °C for 3 min; followed by 35 cycles of 94 °C for 60 s; 42 °C for 90 s of annealing; and 72 °C for 90 s of extension; concluding with a final extension at 72 °C for 90 s and cooling at 4 °C. The PCR conditions for the TEF1-α and RPB2 regions included an initial denaturation at 95 °C for 5 min; followed by 35 cycles at 95 °C for 1 min; annealing at 51 °C and 54 °C for 60 s, respectively; 72 °C for 120 s; followed by a final extension at 72 °C for 10 min. The PCR reactions were conducted using a C1000 Touch^®^ Thermal Cycler (Bio-Rad, Hercules, CA, USA). Additionally, SDS-PAGE electrophoresis was performed on a 1.5% agarose gel for 80 min at 70 V to visualize the PCR products under UV light. Following this, the PCR products were purified using the Wizard^®^ Genomic DNA Purification Kit (PROMEGA, Madison, WI, USA). DNA sequencing was carried out at the Biotechnology Institute of the National Autonomous University of Mexico (IBT-UNAM) in Cuernavaca City, Morelos State, Mexico.

### 2.4. Phylogenetic Analysis

The phylogenetic analysis was conducted using sequences obtained from the ITS, TEF1-α, and RPB2 regions. Alignments for each region were created using the NCBI-BLAST tool. All species reported in the Index Fungorum from the *Arthrobotrys* genus were searched for in the NCBI nucleotide database, and those with sequences from all three regions were included. Additionally, some species from the *Dactylellina* and *Drechslerella* genera, also with three reported regions in the NCBI database, were added to the analysis. In total, 39 species of *Arthrobotrys*, 5 of *Dactylellina*, and 4 of *Drechslerella* were included, with *Vermispora fusarina* (strain YXJ02-13-5) serving as the outgroup species (see Table 1). Multiple alignments for each region were performed using the CLUSTAL algorithm within MEGA Software (v11.0.13). BioEdit (v7.2.5) and MEGA software were utilized to edit and link the three alignments. The best substitution model was determined using jModelTest (v2.1.10), based on the Akaike Information Criterion (AIC). The optimal models were TIM2e + G4 for ITS and SYM + I + G4 for TEF1-α and RPB2. Maximum likelihood analysis was conducted using IQTREE software (v1.6.12), with statistical bootstrap support values obtained through ultrafast bootstrapping with 10,000 replicates. Additionally, Bayesian inference analysis was performed using MrBayes software (v3.2.6). The dataset was partitioned, and equivalent substitution models were applied. Four simultaneous Markov chains were run for 10,000,000 generations, with trees sampled every 1000 generations. The final 90% of the trees were used to estimate the posterior probabilities for the consensus tree. FigTree software (v1.4.4) was used to visualize the tree, while Photoshop CS6 (Adobe Systems, San Jose, CA, USA) and Microsoft PowerPoint (2016) were employed to edit the obtained consensus tree.

### 2.5. Predatory Activity Assessment

Water agar plates (2%) measuring 60 × 15 mm (n = 4) were inoculated with the isolate and maintained at a temperature of 25–30 °C for 10 days. A control series of plates, containing the same sample size and medium but without any fungus, was also prepared. After the 10-day incubation period, each plate from both groups (with and without fungus) received 50 mL of an aqueous suspension containing 200 infective larvae (L_3_) of *Haemonchus contortus*. All plates were incubated at the same temperature for an additional 12 days. The surface of each plate was examined under a stereomicroscope (Leitz, Wetzlar, Germany) on days 2, 7, and 9 to observe trap formation and the nematodes captured. On the 12th day of incubation, the entire content of each plate containing both fungus and captured/non-captured larvae was transferred to a Baermann funnel system. This system facilitated the descent of non-captured larvae from plates with fungus and the descent of all larvae from the control group. The contents remained in the Baermann funnel for 24 h at room temperature (25–30 °C). This technique enabled the recovery of larvae from both the treated and control series, allowing for a comparative analysis of the larvae recovered from each group. It is important to note that the mean number of larvae recovered from the control plates is considered to represent 100% of the initially deposited larvae, unaffected by the fungus. Counting of the recovered larvae was conducted by taking 10–50 µL aliquots, and the number of larvae in these aliquots was counted using an optical microscope (Leica Microsystems, Wetzlar, Germany) with a 10× objective. The entire experiment was performed in triplicate. The larval reduction percentage in the treated group, relative to the control group, attributed to the predatory effect of the fungus, was calculated using the following formula:$$Larval\;reduction\;(\%)=\frac{MRLc-MRLt}{MRLc}*100$$
where

*MRLc* = mean of recovered larvae from the control group *(without fungus)*; 

*MRLt* = mean of recovered larvae from the treated group *(fungus–nematode interaction)*.

### 2.6. Production of the Fungus in Liquid Media

The fungus was grown in two different media: (1) Sweet Potato Dextrose Broth (SPDB); and (2) Czapek–Dox Broth (CzDoxB). The media were prepared in three 250 mL Erlenmeyer flasks, each containing 100 mL of sterile liquid medium. From a 7-day-old culture of the fungus on a 2% water agar plate, three agar cylinders (1 cm diameter x 1 cm thickness) were added to each flask. Control flasks containing the same media without fungus were also prepared. All flasks were incubated at room temperature (25–30 °C) for 21 days under static conditions.

### 2.7. Obtaining Fungal Liquid Culture Filtrates

After incubation, the fungal biomass was separated by filtration to obtain the fungal liquid culture filtrate (LCF) for assessment. The process involved passing the entire content of the flasks through three different types of filters set on a funnel. First, a coffee filter was used for pre-filtration to separate the thick mycelial particles. The recovered liquid was then centrifuged at 3500 rpm for 10 min. The resulting liquid was further filtered through Whatman filter paper No. 4 (25 μm), followed by sterilization using two different syringe filters of 1.1 and 0.22 μm (Millipore, Merck KGaA, Darmstadt, Germany). This filtration process ensured the collection of a biomass-free and sterile LCF. The liquid was then concentrated using a rotary evaporator (Büchi R-300, Flawil, Switzerland) and finally dried by lyophilization (Labconco, Kansas, MO, USA), following the technique described by Pérez-Anzúrez et al. [23].

### 2.8. In Vitro Assessment of Nematocidal Activity of Liquid Culture Filtrates Against Haemonchus contortus Infective Larvae (L_3_)

The in vitro assessment of nematodes exposed to LCF was conducted using 96-well microtiter plates (n = 4). Fifty microliters of the prepared LCF (either from CzDoxB or SPDB) was added to each well, along with 50 μL of an aqueous suspension containing 150 H. contortus L_3_ larvae. Four wells received only sterile distilled water, CzDoxB, or SPDB as negative controls. Additionally, four wells containing 0.5% ivermectin with the nematodes were included as positive controls. Three different concentrations of LCF were tested: 25, 50, and 100 mg/mL. The nematocidal activity of the fungal LCF was assessed after 72 h. The total number of live and dead larvae in each treatment was counted, allowing for the calculation of the mortality percentage attributed to the LCF. The entire experiment was performed in triplicate.

### 2.9. Statistical Analysis

Data from the predation assay were analyzed by comparing the treated and control groups using Student’s *t*-test, with the mean number of recovered larvae considered as the dependent variable (*p* < 0.05). For the analysis of the nematocidal activity of LCF, the Kruskal–Wallis non-parametric method was employed. The statistical software used for these analyses was SPSS version 25 (IBM Corp., New York, NY, USA), with a significance set at (*p* < 0.05).

## 3. Results

### 3.1. Morphological Taxonomic Identification

The isolate has been deposited at the National Center of Genetic Resources (CM-CNRG) in Mexico, which is affiliated with the World Federation of Culture Collections (WFCC No. 1006) and the Latin American Federation for Culture Collections (FELACC No. 56). The deposit code for this sample is CM-CNRG 967. Sequences were uploaded to the NCBI GenBank under the following accession numbers: PQ655528 (TEF), PQ649538 (ITS), and PQ661202 (RPB2). Etymology: the species named “*mendozadegivensis*” is derived from the surname of its discoverer.

Materials examined: The specimen was collected in Coatlán del Río Municipality, State of Morelos, Mexico; the geographic coordinates are 18°45′5″ north latitude and 99°26′8″ west longitude. The specimen was obtained from decaying wood on 23 November 2022 by P. Mendoza de Gives. It is preserved in the fungal collection of the Laboratory of Helminthology at the National Center for Disciplinary Research in Animal Health and Innocuity, which is part of the National Institute of Research in Forestry, Agriculture, and Livestock in Mexico.

#### 3.1.1. Macroscopic Characteristics

The main macroscopic characteristics of the fungus grown on potato dextrose agar (PDA) plates are illustrated in Figure 3A,B. After two weeks, the fungus displayed cotton-like, whitish, and radially scattered mycelial growth (as seen from above in Figure 3A and from below in Figure 3B). In water agar plates, the fungus of the same age exhibited a tiny, whitish, and transparent growth with scattered mycelia.

Likewise, in Figure 3C–H, a series of six microphotographs illustrate various aspects of conidia and conidiophores (C,F); one photo features a single conidium (D). Additionally, four microphotographs taken with a stereomicroscope depict the growth of *A. mendozadegivensis* after 15 days at room temperature (18–25 °C) on corn meal agar plates (I–L). Figure 3L shows *P. redivivus* trapped in three-dimensional adhesive nets on the surface of a water agar plate. Brown catenulated chlamydospores were produced in water agar and PDA cultures that were older than four weeks (Figure 4E,F).

#### 3.1.2. Microscopic Fungal Characteristics

A microscopic photographic set of the main morphological characteristics of the fungus is shown in Figure 3. Additionally, a schematic drawing of the morphological details of the fungus is shown in Figure 4. The fungus produces an extensive net of thin hyphae growing on the agar surface where different kinds of conidiophores develop, i.e., two or three conidiophore stems emerging from the same initial site (bundled) or conidiophores with only a unique stem (Figure 4D(1–3)). The fungus also produces aerial thickened hyphae that act as “mother hyphae” or “major hyphae” (synnematous) where single (non-branched) conidiophores emerge (Figure 4(E1,D5,F2). Mother hyphae can be produced from the agar surface (D4), aerial hypha (F1) and from the nematode body (E1). This fungus produces three-dimensional adhesive nets (Figure 4C(1–3); Figure 4C) where nematodes are trapped (Figure 4E(3,4,7)). Synnematous hyphae are able to produce a large number of single-stem conidiophores along the hyphae (Figure 4E(1,2),F2)). Conidiophores’ lengths range between 31.66 and 109.33 μm. In the apical part of conidiophores, conidia from obovoid to ellipsoidal were produced and they form clusters from 4 to 6 conidia (Figure 4A,B,D). Most of the conidia showed one septum; however, scarce conidia—either non-septate or with the presence of two septa—were also found (Figure 4A). Conidia length ranged from 8.85 to 18.79 μm with a 3.27 to 5.97 μm width.

### 3.2. Molecular Identification

The analysis of the phylogenetic tree—which compared 48 species from nematode-trapping fungi of the Orbiliaceae family with our new isolate—indicated that the closest sequences corresponded to the species *Arthrobotrys polycephala* (strain 1.006). This phylogenetic tree suggested that our new isolate belongs to the genus *Arthrobotrys*; however, none of the closest species previously reported in the NCBI database matched this isolate (see Figure 5).

The DNA sequence analysis enabled us to generate a phylogenetic tree using both Maximum likelihood and Bayesian inference methods, as illustrated in Figure 5. Following morphological analysis, the authors of this study found no previously reported nematode-trapping fungus isolate that exhibited the characteristics of our isolate. Similarly, after conducting molecular analysis and comparing sequences with those of the closest matches in the NCBI database, we concluded that this isolate represents a new species of *Arthrobotrys*, which we propose to name *Arthrobotrys mendozadegivensis*.

After analyzing all the morphometric data, we were unable to achieve a conclusive classification for this isolate; therefore, we decided to support our morphological taxonomic identification with molecular analyses to clarify the species of our isolate. Once we received the sequence from the Institute of Biotechnology of the National Autonomous University of Mexico, we aligned it using the BLASTn tool and analyzed it by comparing it with closely related sequences from the NCBI database. The similarity and coverage percentages, along with the corresponding NCBI accession numbers, are presented in Table 2.

### 3.3. Predatory Activity Assessment

During the incubation process of the plates containing the interaction between fungus and nematodes, several images were captured to illustrate the formation of trapping devices and the nematodes that were trapped (see Figure 6). The analysis of the recovered larvae after the interaction between the fungus and nematodes revealed an average of 28 ± 25.01 larvae recovered from the treated group compared to 125.18 ± 39.84 larvae from the control group. By applying Abbott’s formula, we observed a larval reduction of 76.92%, attributed to the predatory effect of the fungus.

### 3.4. Nematocidal Activity of Liquid Culture Filtrates of Arthrobotrys Mendozadegivensis Against Haemonchus Contortus Infective Larvae (L3) Under In Vitro Conditions

Table 3 displays the mean number of dead larvae recovered from the plates containing the interaction of nematodes with the liquid culture filtrate (LCF) compared to the control plates, along with their standard deviation. The highest mortality rate was observed at the highest concentration of LCF (100 mg/mL).

## 4. Discussion

It is noteworthy that, from the initial microscopic observations conducted with a stereomicroscope on water agar plates sprinkled with decaying wood, we observed an unusual aerial growth at 2X magnification. The growth of mycelia and conidiophores resembled small bushes supporting numerous conidial agglomerations (Figure 3I–K). When we examined these structures at a higher magnification (5×), we noticed that conidiophores emerged either from a “mother” or “guide” hyphae. Additionally, some conidiophores appeared to originate from the same point, or bundle, within the corpses of invaded nematodes (Figure 4E(3,5)). Interestingly, although this isolate produces three-dimensional adhesive nets similar to various species within the genus *Arthrobotrys*, such as *A. oligospora*, *A. arthrobotryoides*, and *A. robusta*, we found that the conidia of our isolate were considerably smaller and distinctly different from those of the mentioned species. For instance, the conidia of *A. oligospora* are pyriform, whereas those of our isolate are elongated and very small, measuring between 8.85 and 18.79 × 3.27 and 5.97 μm. In contrast, the conidia of *A. oligospora* are nearly double the size: 17–35 × 8.5–16 μm. These growth characteristics are not commonly found in other species of nematode-trapping fungi, contributing to the challenges we faced in classifying our isolate. Moreover, when comparing the morphological traits of our new isolate with those documented in the literature, we found some similarities with the species *A. dendroides*. This species is characterized by a similar growth pattern of agglomerated conidiophores resembling bushes. However, *A. dendroides* produces geniculated conidia, which were not observed in our new species. Additionally, our isolate exhibited conidiophores emerging from a single hypha and we noted the presence of catenulated chlamydospores, while *A. dendroides* does not produce chlamydospores.

After analyzing the sequences and comparing them with those previously reported in the NCBI database, we discovered that one uncultured fungus (NCBI accession number: GU053870.1), uploaded by Frohlich et al. [32], exhibited the highest similarity percentage at 99.11%. Unfortunately, information regarding this isolate is incomplete, and it appears that it has not been published or cultured. Additionally, we found three species of nematode-trapping fungi—*Arthrobotrys polycephala*, *A. dendroides*, and *A. ellipsospora*—showing similarity percentages ranging from 91.55% to 93.12%. Specifically, the sequences with the highest similarities were *A. dendroides* at 91.55% (NR_159642.1), *A. ellipsospora* at 91.55% (LC146721.1), and *A. polycephala* at 93.12% (NR160072.1). These species are morphologically distinct from our new isolate. This study marks the first record of *Arthrobotrys mendozadegivensis* worldwide. This species demonstrated significant predatory activity against the blood-feeding nematode parasite of small ruminants, *Haemonchus contortus*, achieving rates close to 77%. This result aligns with the standard predatory activity observed in many *Arthrobotrys* species under in vitro conditions against gastrointestinal parasitic nematodes, such as *H. contortus*, which typically ranges from 70% to 99% (see Table 4). The predatory activity exhibited by this new species of *Arthrobotrys* positions it as a promising candidate for future research as a biological control agent against parasitic nematodes.

This is the first report of the nematocidal activity of the liquid culture filtrates (LCF) from *Arthrobotrys mendozadegivensis* grown in liquid media. After analyzing the results on the nematocidal activity of the LCF produced in SPDB and CzDoxB after 72 h, we observed mortality percentages of 40.90% and 34.91%, respectively. It is worthwhile to compare these results with other studies involving different *Arthrobotrys* species and the lethal activities of their LCF in the same liquid media. For instance, a recent study with *A. musiformis* showed that after growing in CzDoxB, its LCF caused 93.42% larval mortality against *H. contortus*; however, when the same fungus was grown in SPDB, it achieved only 26.80% mortality at its highest concentration of 100 mg/mL [23]. Additionally, another isolate of *A. musiformis* demonstrated 69.96% and 49.84% larval mortality in CzDoxB and potato dextrose broth, respectively [34]. Similarly, another species, *A. sinensis*, grown in a specific medium containing glucose, yeast extract, K_2_HPO_4_, MgSO_4_, ZnSO_4_, FeSO_4_, and CuSO_4_ showed a 64% larval mortality rate against *Angiostrongylus vasorum*, a parasite affecting dogs [39]. In another study, organic extracts from the mycelia of three *Arthrobotrys* species *A. oligospora*, *A. conoides*, and *A. arthrobotryoides* resulted in 72.1%, 81.6%, and 89.51% larval mortality against *H. contortus*, respectively [38]. Comparing our results with those from other species and isolates suggests that different species and isolates, even within the same genus, exhibit varying levels of nematocidal compound production. This variability is influenced by their biochemical and physiological states, which are affected by their microenvironments. This study provides insights into the potential biotechnological applications of these microorganisms for controlling one of the most pathogenic parasitic nematodes that affect small ruminants worldwide. Additionally, this research presents information about a previously unreported species of nematophagous fungus, which is important for systematics and offers potential biotechnological solutions for combating nematodes in the livestock industry.

## Figures and Tables

**Figure 1 jof-10-00888-f001:**
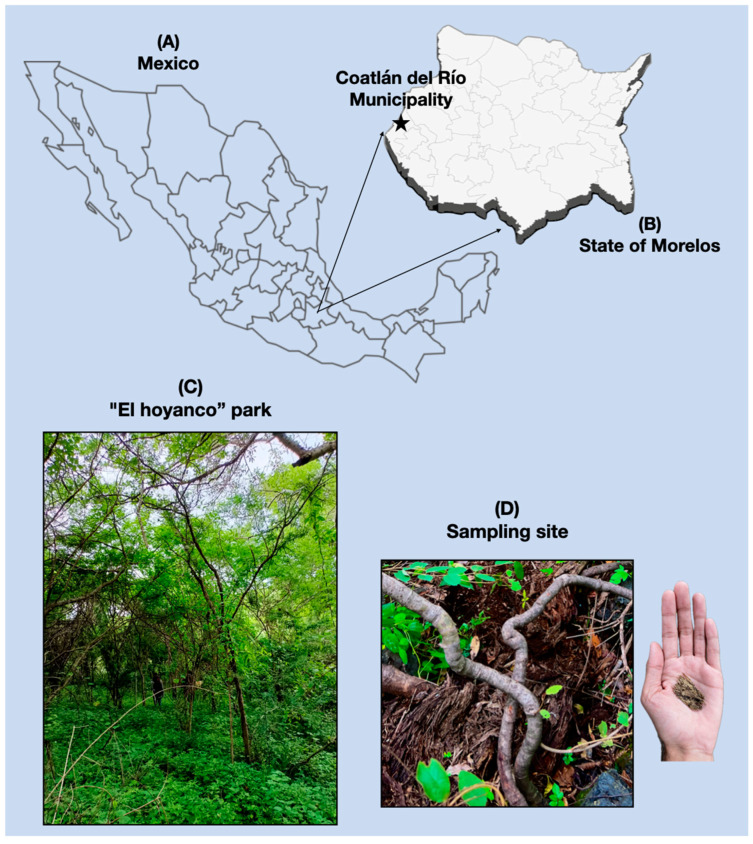
(**A**) Map of the Mexican Republic; (**B**) State of Morelos; (Star) Coatlán del Río Municipality; and (**C**) a picture of the site of sampling at the Eco-touristic park El Hoyanco; (**D**) sampling site.

**Figure 2 jof-10-00888-f002:**
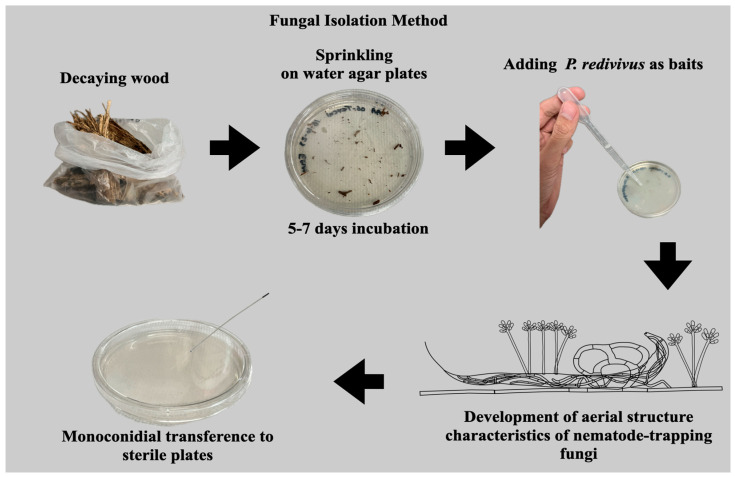
Representative scheme of the nematophagous fungi isolation process from decaying wood.

**Figure 3 jof-10-00888-f003:**
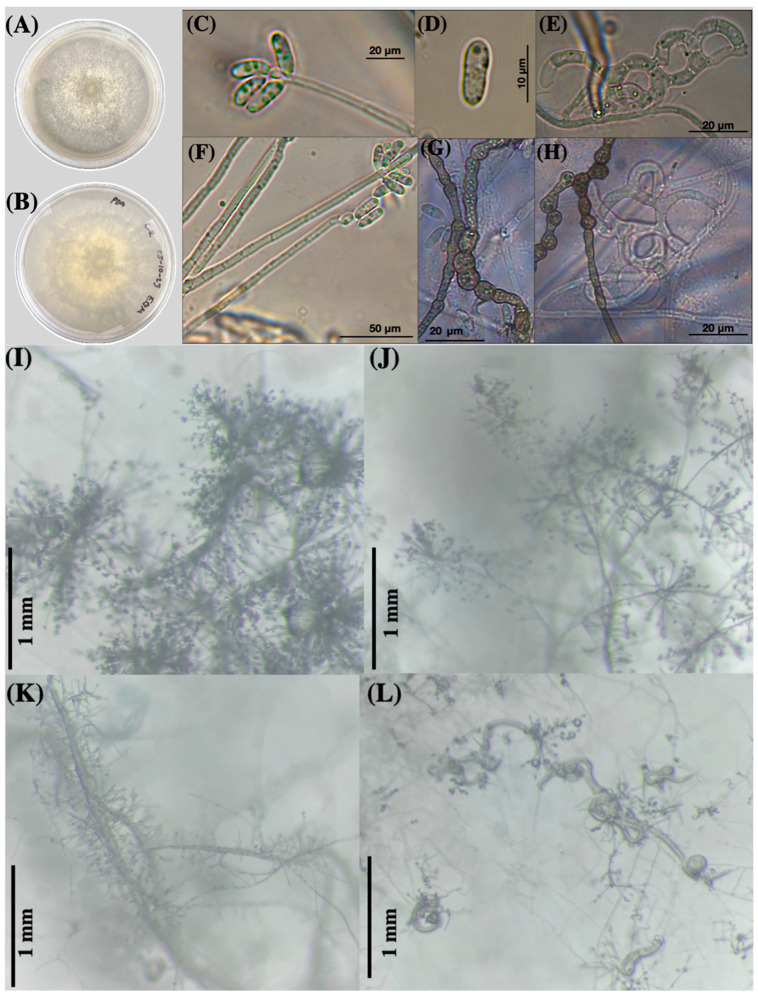
Macroscopic view (**A**,**B**), microphotographs (from a compound microscope, (**C**–**H**)), and views from a stereomicroscope (**I**–**L**) of Potato Dextrose Agar plates of the nematode-trapping fungus *Arthrobotrys mendozadegivensis*; aspect of free-living nematodes of *Panagrellus redivivus* species captured by three-dimensional adhesive nets of the fungus on the water agar surface (**L**); aspect of three-dimensional adhesive nets (**E**,**H**); and catenulated chlamydospores (**G**,**H**).

**Figure 4 jof-10-00888-f004:**
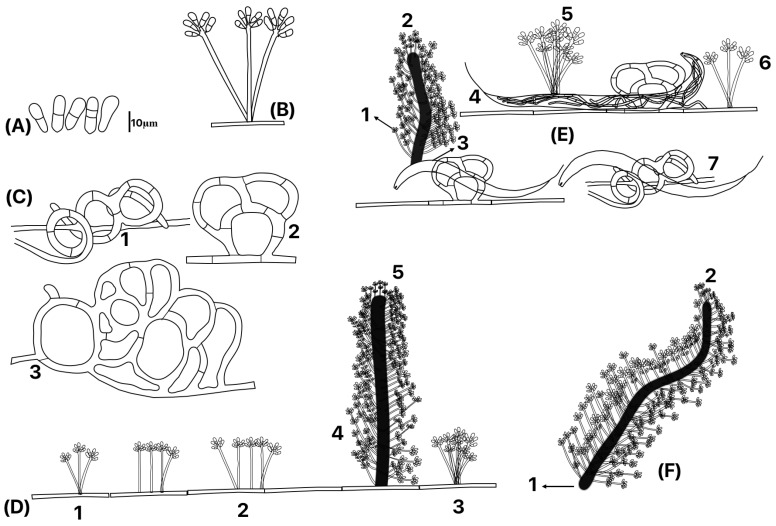
Representative scheme of the development of aerial structures of the nematode-trapping fungus *Arthrobotrys mendozadegivensis*: (**A**) conidia; (**B**) hyphae-produced conidiophores from tillers; (**C**(**1**–**3**)) three-dimensional adhesive nets; (**D**) different kinds of conidiophores emerging from a hypha, (**D1**) three conidiophores emerging from same point from the hypha; (**D2**) individual and bundled conidiophores; (**D3**) bundled conidiophores; (**D4**) Synnematous hypha; (**D5**) conidiophores emerging from a synnematous hypha; (**E**(**1,2**)) conidiophores emerging from a synnematous hypha which in turn emerge from a trapped nematode; (**E3**) trapped nematode and a synnematous hypha emerging from its body; (**E4**) a nematode trapped and invaded by mycelia of the nematode-trapping fungus; (**E5**) bundled conidiophore emerging from the nematode body; (**E6**) a conidiophore growing from a hypha near to the trapped nematode; (**E7**) nematode trapped in a three-dimensional adhesive net; (**F1**) synnematous hypha; and (**F2**) a large amount of single-stem conidiophores emerging from synnematous hypha.

**Figure 5 jof-10-00888-f005:**
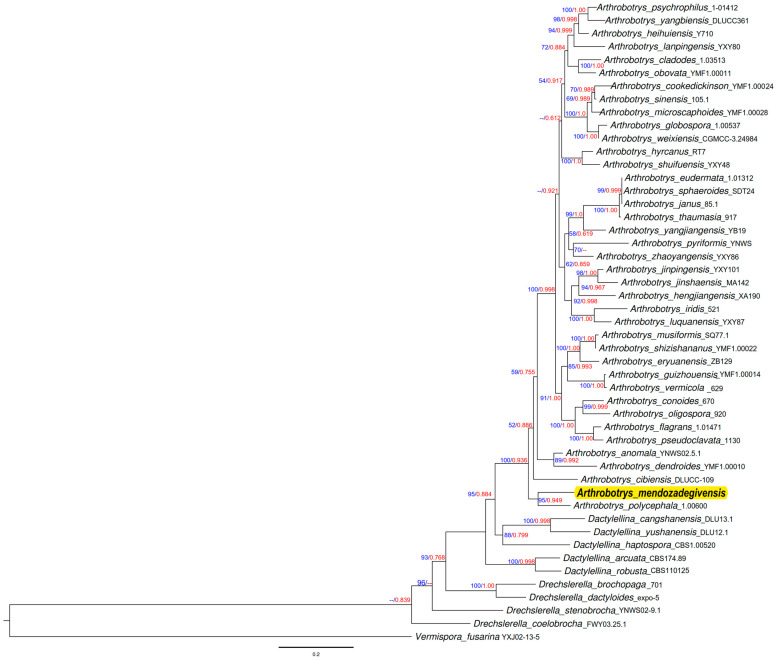
Maximum likelihood tree based on a combined ITS, TEF1-α, and RPB2 sequences from 48 species of Orbiliaceae nematode-trapping fungi. Bootstrap support values for ML greater than 50% and BI posterior probabilities values greater than 0.70 are indicated aside each node (in blue). The new isolate is in red. The tree is rooted by *Vermispora fusarina* YXJ-13-5.

**Figure 6 jof-10-00888-f006:**
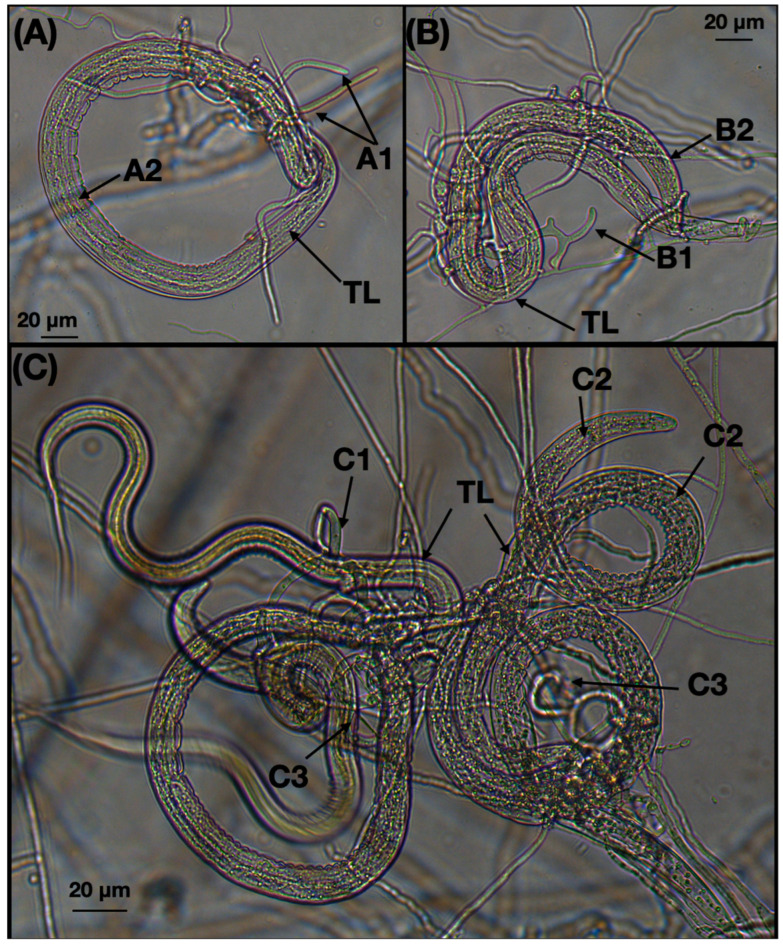
Photographs taken from a Leica Digital Microscope showing *Haemonchus contortus* infective larvae trapped in the three-dimensional adhesive nets (**A**–**C**). (**TL**) trapped larva; (**A1**) hyphae emerging from a trapped nematode body; (**A2**,**B2**,**C2**) invasive hyphae colonizing the intestinal tissues of larvae; (**B1**,**C1**) developing adhesive nets; (**C3**) three-dimensional adhesive nets.

**Table 1 jof-10-00888-t001:** List of the strain sequences from ITS, TEF, and RPB2 regions reported at the NCBI database, used to construct the Maximum likelihood and Bayesian inference phylogenetic tree.

Species	Strain Number	GenBank Accession Number			Reference
		ITS	TEF	RPB2	
*Arthrobotrys anomala*	YNWS02-5-1	AY773451	AY773393	AY773422.1	[25]
*Arthrobotrys cibiensis*	DLUCC 109	OR880379	OR882792	OR882797.1	[26]
*Arthrobotrys cladodes*	1.03513	MH179792	MH179615	MH179892.1	[27]
*Arthrobotrys conoides*	670	AY773455	AY773397	AY773426.1	[25]
*Arthrobotrys cookedickinson*	YMF1.00024	MF948393	MF948550	MF948474.1	[28]
*Arthrobotrys dendroides*	YMF1.00010	MF948388	MF948545	MF948469.1	[28]
*Arthrobotrys eryuanensis*	ZB129	MT612105	OM850307	OM850301	[16]
*Arthrobotrys eudermata*	1.01312	MH179725	MH179576	MH179830.1	[27]
*Arthrobotrys flagrans*	1.01471	MH179741	MH179583	MH179845.1	[27]
*Arthrobotrys globospora*	1.00537	MH179706	MH179562	MH179814.1	[27]
*Arthrobotrys guizhouensis*	YMF1.00014	MF948390	MF948547	MF948471.1	[28]
*Arthrobotrys heihuiensis*	Y710	OR902194	OR882786	OR882801.1	[16]
*Arthrobotrys hengjiangensis*	XA190	OQ946586	OQ989311	OQ989301	[29]
*Arthrobotrys hyrcanus*	RT7	MH367058	OP351540	**___________**	[30]
*Arthrobotrys iridis*	512	AY773452	AY773394	AY773423.1	[25]
*Arthrobotrys janus*	85-1	AY773459	AY773401	AY773430.1.	[25]
*Arthrobotrys jinpingensis*	YXY101	ON808621	ON809552	OR882804.1	[16]
*Arthrobotrys jinshaensis*	MA142	OR902197	OR882789	OR882804	[26]
*Arthrobotrys lanpingensis*	YXY80	ON808618	ON809549	ON809555	[16]
*Arthrobotrys luquanensis*	YXY87	ON808619	ON809550	ON809556	[16]
*Arthrobotrys microscaphoides*	YMF1.00028	MF948395	MF948552	MF948476.1	[28]
*Arthrobotrys musiformis*	SQ77-1	Y773469	AY773411	AY773440.1	[25]
*Arthrobotrys obovata*	YMF1.00011	MF948389	MF948546	MF948470.1	[28]
*Arthrobotrys oligospora*	920	AY773404	AY773462	AY773433.1	[25]
*Arthrobotrys polycephala*	1.006	MH179724	MH179575	MH179829.1	[27]
*Arthrobotrys pseudoclavata*	1130	AY773446	AY773388	AY773417.1	[25]
*Arthrobotrys psychrophila*	1.01412	MH179727	MH179578	MH179832.1	[27]
*Arthrobotrys pyriformis*	YNWS02-3-1	AY773450	AY773392	*AY773421.1*	[25]
*Arthrobotrys shizishananus*	YMF1.00022	MF948392	MF948549	MF948473.1	[28]
*Arthrobotrys shuifuensis*	YXY48	ON808617	ON809548	ON809554.1.	[16]
*Arthrobotrys sinensis*	105-1	AY773445	AY773387	AY773416.1	[25]
*Arthrobotrys sphaeroides*	SDT24	AY773465	AY773407	AY773436.1	[25]
*Arthrobotrys thaumasia*	917	AY773461	AY773403	AY773432.1.	[25]
*Arthrobotrys vermicola*	629	AY773454	AY773396	AY773425.1	[25]
*Arthrobotrys weixiensis*	CGMCC 3.24984	OQ946585	OQ989310	OQ989300.1	[29]
*Arthrobotrys yangbiensis*	DLUCC 36-1	OR880382	OR882795	OR882800.1	[26]
*Arthrobotrys yangjiangensis*	YB19	OR902196	OR882788	OR882803.1	[26]
*Arthrobotrys zhaoyangensis*	YXY86	ON808620	ON809551	ON809557	[16]
*Dactylellina arcuate*	CBS 174.89	AF106527	DQ999852	DQ999799.1	[25]
*Dactylellina cangshanensis*	DLU 13-1	MK372062	MN915115	MN915114.1	[31]
*Dactylellina haptospora*	CBS 100520	DQ999820	DQ999850	DQ999814.1	[25]
*Dactylellina robusta*	CBS 110125	DQ999821	DQ999851	DQ999800.1	[25]
*Dactylellina yushanensis*	DLU12-1	MK372061	MN915113	MN915112.1	[31]
*Drechslerella brochopaga*	701	AY773456.1	AY773398.1	AY773427.1	[25]
*Drechslerella coelobrocha*	FWY03-25-1	AY773464.1	AY773406.1	AY773435.1	[25]
*Drechslerella dactyloides*	Expo-5	AY773463.1	AY773405.1	AY773434.1	[25]
*Drechslerella stenobrocha*	YNWS02-9-1	AY773460.1	AY773402.1	AY773431.1	[25]
*Dactylaria* sp.	YNWS02-7-1	AY773457	AY773399	AY773428.1	[25]
*Vermispora fusarina*	YXJ02-13-5	AY773447	AY773418	AY773418.1	[25]
*Arthrobotrys* *mendozadegivensis*	INIFAP-EGM-01	PQ649538	PQ655528	PQ661202	Present study

**Table 2 jof-10-00888-t002:** Similarity and coverage percentages of the sequence of a nematode-trapping fungus isolate after comparison with reported sequences from the GenBank–NCBI database, using the partial sequence of ITS1, 5.8S, and ITS2 regions.

Fungus	Query Cover %	Identity %	GenBank Accession Number
Uncultured fungus	98	99.11	GU053870.1
*Arthrobotrys polycephala*	98	93.12	NR_160072
*A. polycephala*	98	93.12	MH855875.1
*A. dendroides*	98	91.55	NR_159642.
*A. dendroides*	98	91.55	MH861894.1
*A. ellipsospora*	98	91.55	LC146721.1

**Table 3 jof-10-00888-t003:** Mean of dead and live *Haemonchus contortus* infective larvae recovered from the wells of microtiter plates after 72-h interaction with *Arthrobotrys mendozadegivensis* culture filtrates and mortality percentages.

Media	Concentration	Dead Larvae ± SD	Total Larvae ± SD	Mortality (%) ± SD
CzDoxB	PBS	1.92 ± 1.46	101.24 ± 11.28	1.9 ± 1.68
	NF	4.10 ± 2.23	95.03 ± 19.20	4.31 ± 3.41
	25	4.31 ± 2.50	103.65 ± 10.65	4.15 ± 2.43
	50	7.45 ± 8.56	100.09 ± 13.00	7.44 ± 8.68
	100	33.34 ± 15.49	95.49 ± 14.41	34.91 ± 16.67
SPDB	PBS	1.05 ± 0.63	77.23 ± 17.49	1.36 ± 0.35
	NF	3.37 ± 2.5	97.25 ± 15.31	3.47 ± 2.07
	25	2.23 ± 1.21	92.66 ± 25.21	2.41 ± 1.82
	50	2.51 ± 3.39	92.73 ± 19.24	2.71 ± 2.62
	100	35.25 ± 16.55	86.18 ± 8.33	40.90 ± 18.66

CzDoxB = Czapek–Dox Broth; SPDB = Sweet Potato Dextrose Broth; PBS = phosphate buffer solution (pH = 7.2); NF = liquid culture filtrates without fungus.

**Table 4 jof-10-00888-t004:** Trapping ability of different isolates of *Arthrobotrys* species under in vitro conditions against *Haemonchus contortus* infective larvae and other parasitic nematodes.

Arthrobotrys Species	Blank Nematode	Predatory Activity	Author
*Arthrobotrys oligospora*	*Haemonchus contortus*	76–79%	[33]
*A. musiformis*	*H. contortus*	74.9%	[34]
*A. cladodes*	*Strongyloidess papillosus*	99.5%	[35]
*A. oligospora*	*Trichostrongylus colubriformis* *H. contortus*	90–99.99%	[36]
*A. oligospora* *A. flagrans*	*H. contortus*	47.5% 41.8%	[37]
*A. conoides*	*H. contortus*	75%	[38]
*A. mendozadegivensis*	*H. contortus*	76.92%	(Present study)

## Data Availability

The original contributions presented in the study are included in the article, further inquiries can be directed to the corresponding author.

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
