# Peer review of "Arthrobotrys mendozadegivensis sp. nov. (Fungi: Orbiliales) from Mexico: Predatory Activity and Nematocidal Activity of Its Liquid Culture Filtrates Against Haemonchus contortus (Nematoda: Trichostrongylidae)"

_jof, 2024, doi:10.3390/jof10120888_

Round 1
Reviewer 1 Report
This study primarily focuses on the isolation, identification, and assessment of a new nematode-trapping fungus (NFT) from decaying wood in Morelos State, Mexico. The fungus, named Arthrobotrys mendozadegivensis, exhibited a predatory activity of 76.92%, and its liquid culture filtrates were effective in killing Haemonchus contortus larvae. The study is highly significant, and the author's work in isolation and identification is thorough. However, it appears that many of the results have been included in the discussion section, which may cause confusion for readers. The results section should objectively present the research findings while the discussion section is meant for interpreting those findings and comparing them to previous studies. It is suggested that the author review both sections and make appropriate revisions.
My specific comments are focused on minor technical mistakes and I do not have any bad criticism about the design or structure of the study.
Line 42: even the Antarctic [9].
Line 99-100: A suggestion: In describing the process of fungal cultivation, it can be emphasized that sterile operations are used, and that the water added to the bottom of the Petri dish was sterile water.
Line 172: why you inoculated the isolate under dark conditions?
Line 190: How many parallel groups did you conduct in your experiment? How many experiments were repeated in total?
Line 228: Was there a parallel group set up during the experiment, or was it just one set of experiments repeated three times?
Line 232: the 'p' needs to be italicized. Please read the full text, all p-values indicating statistical significance need to be italicized.
Line 242: change “name” to “named”.
Line 373: The content of Table 3 seems more suitable to be included in the results, as it is evidence that you have identified the isolated fungi as newly discovered fungi.
Line 381-382: The part describing the results of Table 3 seems more suitable to be included in the results.
Line 387: the predatory activity you showed in section 3.3 was 76.92%.
Line 395: Why is section 3.4 after section 4? The sub-heading of section 3.4 is repeated from the previous section 3.4.
Line 398-400: This sentence and Table 4 seem more suitable to be placed in the Results section, as you used the data from Table 4 in Section 3.3.
Author Response
--Line 42: even the Antarctic [9].
Authors: The dot was moved after brackets.
--Line 99-100: A suggestion: In describing the process of fungal cultivation, it can be emphasized that sterile operations are used and that the water added to the bottom of the Petri dish was sterile water.
Authors: In the section about Fungal Isolation we state that the whole procedure was performed under sterile conditions (Lines 90-96). We have also inserted the term “sterile” when we refer to water used to prepare the humid chamber (line 104).
--Line 172: why you inoculated the isolate under dark conditions?
Authors: We´re sorry! We certainly did not use dark conditions! This was written by mistake. We deleted this big error!
--Line 190: How many parallel groups did you conduct in your experiment? How many experiments were repeated in total?
Authors: We actually used two groups one control (without fungus) and other treated (With fungus). Each experiment considered 4 plates (n=4); The entire experiment was performed in triplicate, giving a total of 12 plates per group.
--Line 228: Was there a parallel group set up during the experiment, or was it just one set of experiments repeated three times?
Author: We conducted three experiments using 4 plates for each group.
--Line 232: the 'p' needs to be italicized. Please read the full text, all p-values indicating statistical significance need to be italicized.
Authors: The letter “p” was italized.
--Line 242: change “name” to “named”.
Authors: this term was corrected
--Line 373: The content of Table 3 seems more suitable to be included in the results, as it is evidence that you have identified the isolated fungi as newly discovered fungi.
Authors: We agree with your comment!
We have moved Table 3 to the Results section
--Line 381-382: The part describing the results of Table 3 seems more suitable to be included in the results.
Authors: IDEM
--Line 387: the predatory activity you showed in section 3.3 was 76.92%.
Authors: We corrected the predation percentage in the Discussion section.
--Line 395: Why is section 3.4 after section 4? The sub-heading of section 3.4 is repeated from the previous section 3.4.
Authors: Sorry! Big mistake! We have corrected this sequential number.
--Line 398-400: This sentence and Table 4 seem more suitable to be placed in the Results section, as you used the data from Table 4 in Section 3.3.
Authors: We believe that Table 4, which illustrates the predatory activity of other fungal species as reported by various authors for comparison, should remain in the Discussion section. However, if you feel that it would be more appropriate to move it to the Results section, we are open to making that change.

Reviewer 2 Report
The study reported a novel entomopathogenic fungi characterized as Arthrobotrys mendozadegivensis. The strain was characterized based on their phenotypic distinctiveness and molecular characterization based on DNA sequences of multiple loci, which includes comparison with other known fungal strains belonging to the genus Arthrobotrys. The authors reported certain dissimilarities with other known fungal species across the examined physical features; conidiophores, hyphae, conidia. This is in addition to the molecular characterization involving multi-loci regions; ITS, TEF, and RPB2, that revealed significantly differences in comparison to other fungal species that have been deposited in NCBI database. The liquid culture filtrates of the characterized EPF recorded between 35-41% pathogenicity against larvae of Haemonchus contortus, dependent on the culture media used.
The isolation and characterization of novel promising entomopathogenic fungal strains could offer solutions to combating insect pests and pathogen problems, specifically in the area of dealing with induced resistance.
Although there are a few parts of the manuscript that could be further improved. Specifically, the abstract and the discussion need to be extensively revised. Other corrections to be made are only minor issues with the texts and formating.
Overall, the work is innovative, adds value to science and the quality of manuscript is up to standard. In addition, the experiment procedures used in carrying out the studies were well highlighted. I assume that the multi-loci regions adopted for molecular and phylogenetic assessment is sufficient and generally acceptable as to ensure fungal isolates are accurately classified.
The authors need to highlight the paper's innovative contributions. For instance, the abstract could be concluded with a statement summarizing the importance of the paper. What is the significance of this paper, I mean, what does it add to science?
L82 - room temperature (18-25 °C). Also correct in L101
L91 – passes?
L111 – Please provide manufacturer details, and country of production.
L132-134 – 94 °C, 42 °C, 72 °C, 4 °C. Please, correct throughout the manuscript
L167 – Under the reference column, I am wondering if there is need to include the author and year of publication in addition to the cited reference.
Authors are advised to recheck the journal guidelines as to ensure that the correct citation style is used throughout the manuscript.
That being said, while citing citation in-between text, I assume there is no use for the year of publication to be included. For instance, L60 should be “White et al. [18], Stielow et al. [19], and Zhang et al. [16].
See also, L122 – Correct as, “Tigano-Milani et al. [24]” Check all over the manuscript to effect this change. See L216, L377, etc.
Also ensure that the correct table format is used.
L203 - 25-30 °C. See my previous comments.
L219 – In-vitro assessment
L222 – “H. contortus” should be in italic
L235 – Also, “p” in “p < 0.05” also need to italicized
A large part of the discussion section read like they ought to form part of the results. See for instance, L340-363; L395-416. The discussion section needs to be substantially improved.
L394 – Under Author (Country) column – check my previous comments as regards referencing style.
L395-396 – I do not understand the need for the sub-heading here. Perhaps this was copied from L329-330. Please, delete.
Author Response
Reviewer #2:
Major comments
The study reported a novel entomopathogenic fungi characterized as Arthrobotrys mendozadegivensis. The strain was characterized based on their phenotypic distinctiveness and molecular characterization based on DNA sequences of multiple loci, which includes comparison with other known fungal strains belonging to the genus Arthrobotrys. The authors reported certain dissimilarities with other known fungal species across the examined physical features; conidiophores, hyphae, conidia. This is in addition to the molecular characterization involving multi-loci regions; ITS, TEF, and RPB2, that revealed significantly differences in comparison to other fungal species that have been deposited in NCBI database. The liquid culture filtrates of the characterized EPF recorded between 35-41% pathogenicity against larvae of Haemonchus contortus, dependent on the culture media used.
The isolation and characterization of novel promising entomopathogenic fungal strains could offer solutions to combating insect pests and pathogen problems, specifically in the area of dealing with induced resistance.
Although there are a few parts of the manuscript that could be further improved. Specifically, the abstract and the discussion need to be extensively revised. Other corrections to be made are only minor issues with the texts and formating.
Overall, the work is innovative, adds value to science and the quality of manuscript is up to standard. In addition, the experiment procedures used in carrying out the studies were well highlighted. I assume that the multi-loci regions adopted for molecular and phylogenetic assessment is sufficient and generally acceptable as to ensure fungal isolates are accurately classified.
Detail comments
The authors need to highlight the paper's innovative contributions. For instance, the abstract could be concluded with a statement summarizing the importance of the paper. What is the significance of this paper, I mean, what does it add to science?
Authors actions:
The significance and contributions to systematics, as well as the potential applications of this new species, are highlighted in both the abstract and the Discussion section.
L82 - room temperature (18-25 °C). Also correct in L101
Authors actions:
This mistake was corrected.
L91 – passes?
Authors actions:
Transfers
L111 – Please provide manufacturer details, and country of production.
Authors actions:
Free-living nematodes were commercially acquired as living fish food from a local pet store.
L132-134 – 94 °C, 42 °C, 72 °C, 4 °C. Please, correct throughout the manuscript
Authors actions:
Everyone of the mistakes were corrected.
L167 – Under the reference column, I am wondering if there is need to include the author and year of publication in addition to the cited reference.
Authors actions:
I think, authors in this Table are unnecessary! We have deleted them!
Authors are advised to recheck the journal guidelines as to ensure that the correct citation style is used throughout the manuscript.
Authors actions:
The Journal of Fungi guidelines were consulted and our manuscript was adjusted.
That being said, while citing citation in-between text, I assume there is no use for the year of publication to be included. For instance, L60 should be “White et al. [18], Stielow et al. [19], and Zhang et al. [16].
Authors actions:
These errors were corrected.
See also, L122 – Correct as, “Tigano-Milani et al. [24]” Check all over the manuscript to effect this change. See L216, L377, etc.
Authors actions:
These errors were corrected. The whole manuscript was re-checked and corrected.
Also ensure that the correct table format is used.
Authors actions:
These errors were corrected.
L203 - 25-30 °C. See my previous comments.
Authors actions:
L219 – In-vitro assessment
Authors actions:
This change was made.
L222 – “H. contortus” should be in italic
Authors actions:
This mistake was corrected
L235 – Also, “p” in “p < 0.05” also need to italicized
Authors actions:
“p” was italicized
A large part of the discussion section read like they ought to form part of the results. See for instance, L340-363; L395-416. The discussion section needs to be substantially improved.
Authors actions:
One of our reviewers asked us to re-order the paragraphs from the Discussion section to the Results section; so, one part of the discussion section was moved to Results.
L394 – Under Author (Country) column – check my previous comments as regards referencing style.
Authors actions:
The column was corrected according to your kind suggestions.
Authors actions:
L395-396 – I do not understand the need for the sub-heading here. Perhaps this was copied from L329-330. Please, delete.
Author actions:
The sub-heading was removed

Reviewer 3 Report
The authors propose a new species of fungus in the genus Arthrobotrys and they investigated its predatory and nematicidal activity against the animal parasitic nematode Haemonchus contortus. The manuscript is interesting and scientifically sound. The specific gap of the manuscript is that the authors were not provided the sequence accession numbers. In addition, there are some specific comments that should be followed to further improve the manuscript.
Specific comments
· Line 45. „..Orbiliomycetes group…“ Taxonomically, the Orbiliomycetes is a class of fungi, not a group.
· Figure 3. (I-L) Photos are dimmed and detailes are not distinct.
· Figure 5. The probability values are confusing and not distict. Please, make two independent phylogenetic trees with branches and probability values based on ML and MrBayes separately, to obtain more comprehensive trees. In addition, please provide the sequence accession numbers from the official genebank database.
· Figure 6. (C) C2 points toward the nematode not to the invasive hyphae.
· Line 400. „..observed low mortality percentages …“ I would remove the term low, 40% is not cosidered to be a low mortality.
Author Response
The authors propose a new species of fungus in the genus Arthrobotrys and they investigated its predatory and nematicidal activity against the animal parasitic nematode Haemonchus contortus. The manuscript is interesting and scientifically sound. The specific gap of the manuscript is that the authors were not provided the sequence accession numbers. In addition, there are some specific comments that should be followed to further improve the manuscript.
Authors: We uploaded the corresponding sequences using three genes to GenBank just a couple of days back. So far, we have obtained only one accession number:
PQ649538 that corresponds to ITS. The other two accession numbers (Submissions 2898475 (RPB2) and 2898015 (TEF2)) must be sent us, in brief.
As soon as we get them we will notify you!
Line 45. „..Orbiliomycetes group…“ Taxonomically, the Orbiliomycetes is a class of fungi, not a group.
Authors: We have corrected this mistake.
- Figure 3. (I-L) Photos are dimmed and details are not distinct.
Authors: Dear reviewer, we actually capture these images using the stereomicroscope, we tried to improve the quality of these images; however, the so small size of these fungal structures difficult to increase the resolution of these images. For that reason, we included the microphotographs taken with the optical microscope as a complementary image to get a wider view.
- Figure 5. The probability values are confusing and not distict. Please, make two independent phylogenetic trees with branches and probability values based on ML and MrBayes separately, to obtain more comprehensive trees. In addition, please provide the sequence accession numbers from the official genebank database.
Authors: We built a phylogenetic tree using three sequences ITS, RPB2 and TEF2a; because when we tried to analyze every tree independently, we found very similar topologies. In that tree, we included the two ML and MrBayes values separated by a slash; although both values are marked in blue color and this can confuse. In the new version, we have changed the color of these values and we used blue color for ML values and red color for MrBayes values to avoid confusion.
- Figure 6. (C) C2 points toward the nematode not to the invasive hyphae.
Authors: Sorry, you´re right! We tried to point the hyphae inside the nematode body.
In the new version this mistake was corrected.
- Line 400.„..observed low mortality percentages …“ I would remove the term low, 40% is not cosidered to be a low mortality.
Authors: The term “low” was deleted

Round 2
Reviewer 3 Report
The authors have still not provided the sequence accession numbers, except for ITS region. However, they have not included it in the text. I do not know why? If submitting new sequences is complicated, they can get assistance from NCBI. I would support publication if editors can provide a corrigendum afterwords. Otherwise, I would wait until they obtain all accession numbers and include them in the text.
• Lines 424-426. The sentence is repeating sentences given in lines 419-423.
Author Response
Reviewer
The authors have still not provided the sequence accession numbers, except for ITS region. However, they have not included it in the text. I do not know why? If submitting new sequences is complicated, they can get assistance from NCBI. I would support publication if editors can provide a corrigendum afterwords. Otherwise, I would wait until they obtain all accession numbers and include them in the text.
Authors:
We finally obtained the missing NCBI GenBank accession numbers and inserted them in the paragraph, lines 243-245. We also included this information at the end of Table 1.
Reviewer
Lines 424-426. The sentence is repeating sentences given in lines 419-423.
Authors:
We have modified and unified the two sentences in only one.
